# Synthesis, Cytotoxic Evaluation, and Structure-Activity Relationship of Substituted Quinazolinones as Cyclin-Dependent Kinase 9 Inhibitors

**DOI:** 10.3390/molecules28010120

**Published:** 2022-12-23

**Authors:** Hamad M. Alkahtani, Amer Alhaj Zen, Ahmad J. Obaidullah, Mohammed M. Alanazi, Abdulrahman A. Almehizia, Siddique Akber Ansari, Fadilah Sfouq Aleanizy, Fulwah Yahya Alqahtani, Rana M. Aldossari, Raghad Abdullah Algamdi, Lamees S. Al-Rasheed, Sami G. Abdel-Hamided, Alaa A.-M. Abdel-Aziz, Adel S. El-Azab

**Affiliations:** 1Department of Pharmaceutical Chemistry, College of Pharmacy, King Saud University, P.O. Box 2457, Riyadh 11451, Saudi Arabia; 2Chemistry & Forensics Department, Clifton Campus, Nottingham Trent University, Nottingham Ng11 8NS, UK; 3Department of Pharmaceutics, College of Pharmacy, King Saud University, P.O. Box 22452, Riyadh 11495, Saudi Arabia; 4Department of Pharmacology & Toxicology, College of Pharmacy, 11 Prince Sattam Bin Abdulaziz University, P.O. Box 173, Al-Kharj 11942, Saudi Arabia; 5Department of Pharmaceutical Chemistry, Faculty of Pharmacy, Al-Azhar University, Cairo 11884, Egypt

**Keywords:** quinazolinones, cytotoxic agents, CDK9, molecular docking

## Abstract

Cyclin-dependent kinase 9 (CDK9) plays a critical role in transcriptional elongation, through which short-lived antiapoptotic proteins are overexpressed and make cancer cells resistant to apoptosis. Therefore, CDK9 inhibition depletes antiapoptotic proteins, which in turn leads to the reinstatement of apoptosis in cancer cells. Twenty-seven compounds were synthesized, and their CDK9 inhibitory and cytotoxic activities were evaluated. Compounds **7**, **9**, and **25** were the most potent CDK9 inhibitors, with *IC*_50_ values of 0.115, 0.131, and 0.142 μM, respectively. The binding modes of these molecules were studied via molecular docking, which shows that they occupy the adenosine triphosphate binding site of CDK9. Of these three molecules, compound **25** shows good drug-like properties, as it does not violate Lipinski’s rule of five. In addition, this molecule shows promising ligand and lipophilic efficiency values and is an ideal candidate for further optimization.

## 1. Introduction

Cyclin-dependent kinases (CDKs), a family of Ser/Thr PKs belonging to the CMGC [Cyclin-dependent kinases (CDKs), Mitogen-activated protein kinases (MAPKs), Glycogen synthase kinases (GSKs), and Cdc2-like kinases (CLKs)] superfamily, play an essential role in controlling and regulating the cell cycle and the transcription of genes in eukaryotes [1,2,3,4,5,6]. In humans, 20 different CDKs (1–20) have been discovered and can be divided into two main categories according to their primary function. CDK1-4 and CDK6 regulate cell cycle phases [3,4,5,6,7], whereas CDK7-9 and CDK11 promote transcription for cell growth, differentiation, and viral pathogenesis [3,4,5,6,8,9]. CDK9 is overexpressed in various solid and hematological malignancies, where CDK9 is the principal regulator that stimulates transcriptional elongation, through which short-lived antiapoptotic proteins (e.g., Mcl-1) are overexpressed, leading to cancer cells becoming resistant to apoptosis [9]. Pharmacological inhibition of CDK9 induces apoptosis in cancer cells, making it a potential drug target in oncology [10,11,12,13,14,15]. It is expected that selective CDK9 inhibitors will lead to an improvement in the toxicity profile of currently available CDK9 inhibitors [16]. Moreover, unlike other transcriptional CDKs, such as CDK7 and CDK8, CDK9 has been validated as a druggable target for treating other diseases, such as cardiac hypertrophy and HIV infection [9,17]. Several CDK9 inhibitors have been synthesized, with many entering clinical trials. The vast majority of these small molecules are potent competitive inhibitors. In addition, they are effective against hematological malignancies as well as solid tumors such as breast cancer [11,18,19,20,21]. These include, derivatives of flavonoids (flavopiridol, **I**), thiazoles (SNS-032, **II**), pyrimidines (CDKI-73, **III**), pyridines (AZD4573, **IV**), and benzimidazoles (5,6-dichlorobenzimidazole 1-beta-D-ribofuranoside (BDR), **V**) [11,14,15,18,19,22]. Quinazoline derivatives are known for their wide range of therapeutic activities, such as anticancer, antiviral, antihypertensive, and antidiabetic [23,24,25]. This wide range of therapeutic activities exhibited by quinazoline derivatives is mediated by inhibiting several molecular targets, such as protein kinases, carbonic anhydrases, antitumor and cyclooxygenases (COX) [26,27,28,29,30,31,32,33,34,35,36,37,38,39,40,41,42]. In addition, they exhibit promising CDK9 inhibitory activity (such as compound **VI and VII**) that warrants further investigation and improvement (Figure 1) [43,44].

As a continuation of our studies on quinazoline derivatives (**VI**) as promising CDK9 inhibitors that exhibit cytotoxic activity [45], several quinazolin-4-ones linked to substituted anilides and 4-sulfamoylphenethyl were designed, as shown in Figure 2. In addition, the introduction of different substitutions at position 6 of the quinazolin-4-one ring (**VII**) was performed to improve CDK9 inhibitory activity via possible interaction with the gatekeeper region of CDK9. It was expected that such modifications would improve the antiproliferative activity of these molecules against cancer cells by increasing their affinity for the CDK9 enzyme. Molecular docking studies were carried out on this new series of compounds to explore the structural requirements for inhibitory activity toward CDK9.

## 2. Results and Discussion

### 2.1. Chemistry

2-Mercapto-3-(4-sulfamoylphenethyl)quinazolin-4-one (**1**) was prepared, as reported previously [46], in a yield of 95%. Compounds **2**–**16** were prepared in excellent yields (>90%) by treating compound **1** with 2-chloro-*N*-substituted-amide and potassium carbonate at room temperature in acetone, as reported previously in the literature, the reaction scheme of which is shown in Figure 1 [26].

Compounds **17**–**20** and **29** were prepared as reported previously (Figure 3) [31,47,48,49]. 2-(3-Bromobenzamido)-5-methylbenzoic acid (**23**) was prepared in 97% yield via the reaction of 5-methylanthranilic acid (**21**) with 3-bromobenzoyl chloride (**22**) in pyridine at room temperature (Figure 2). Compound **23** underwent hot cyclization in acetic anhydride to give compound **24** in 95% yield.

Multiple spectroscopic techniques were used to confirm the structures of the target compounds **23**–**28**. The chemical structure of compound **23** was established by the presence of the carboxylic (COOH), amide (CONH), and methyl (CH3) peaks at 12.06, 8.51, and 2.33 ppm in ^1^H NMR and 170.52, 163.37 and 20.73 ppm in ^13^C NMR, respectively. 2-(3-Bromophenyl)-6-methyl-4H-benzo[d][1,3]oxazin-4-one (**24**) was confirmed by the disappearance of the carboxylic and amide peaks of compound **23** in NMR spectra. In addition to the occurrence of a new carbonyl group due to 4H-benzo[d][1,3]oxazin-4-one at 159.02 ppm in ^13^C NMR. Boiling compound **24** in formamide gave 2-(3-bromophenyl)-6-methylquinazolin-4(3H)-one (25) in 80% yield. Its formation was confirmed by the disappearance of the carbonyl group of benzo[d][1,3]oxazin-4-one (**24**) at 159.02 ppm and the appearance of a new carbonyl group peak due to the quinazoline nucleus at 165.30 ppm in ^13^C NMR spectrum. 2-(3-Bromophenyl)-6-methylquinazoline-4(3H)-thione (**26**) was prepared in a yield of 68% by boiling compound 25 with phosphorus pentasulfide (P2S5) in toluene. Its successful formation was confirmed by singlet peaks related to the thioamide (CSNH) group at 13.94 ppm in 1H NMR and 187.62 ppm in ^13^C NMR spectra, respectively. Furthermore, 3-amino-2-(3-bromophenyl)-6-methylquinazolin-4(3H)-one (**27**) was obtained in a yield of 87% by heating compound **24** with hydrazine hydrate in ethanol, while 2-(3-bromophenyl)-3-hydroxy-6-methylquinazolin-4(3H)-one (**28**) was obtained in a yield of 84% by heating compound **24** with hydroxylamine hydrochloride in dry pyridine. Compounds **27** and **28** were identified by the presence of new characteristic amine (NH_2_) and hydroxyl (OH) peaks at 5.66 and 11.73 ppm, respectively, in ^1^H NMR spectra, as shown in Figure 2.

### 2.2. Structure-Activity Relationship (SAR) Analysis

The CDK9 inhibition activities of the target compounds were evaluated, with flavopiridol used as a reference. Compound **1** has a similar scaffold to those of several reported CDK9 inhibitors and shows promising CDK9 inhibitory activity with an *IC*_50_ value of 0.644 μM [39]. Therefore, it was selected as a hit compound and modified with different substituents at the quinazoline position 2 to improve its inhibition properties. The results show that introducing acetamide and acetanilide groups at position 2 in compounds **2** and **3** showed better inhibition than the hit compound, with *IC*_50_ values of 0.454 and 0.421 μM, respectively, as shown in Table 1. To investigate the effects that other functional groups substituted at the *p*-position of the acetanilide ring have on improving the potency of the compound, analogs **4**−**8** with various substituents, such as methyl, acetyl, methoxy, and ethoxy groups, were prepared and evaluated. Compound **4**, with a p-methyl group, displays lower inhibitory activity than compounds (**1**–**3**), with an *IC*_50_ value of 0.788 μM, as shown in Table 1.

Furthermore, substituting the *p*-methyl group for a larger group, such as an acetyl group used to generate compound **5**, was found to be detrimental as it was less potent than the previous analogs, showing an *IC*_50_ value of 0.829 μM, as shown in Table 1. However, introducing a methoxy group at the *p*-position gave compound **6**, which led to a recovery in the inhibitory activity, exhibiting an *IC*_50_ value of 0.463 μM. Interestingly, when an ethoxy group was introduced, as in compound **7**, the inhibitory activity increased by around 6-fold compared to the hit compound **1**, with the compound exhibiting an *IC*_50_ value of 0.115 μM. However, replacing the *p*-methoxy group with 3,4,5-trimethoxy groups in the acetanilide ring led to compound **8**, which demonstrated less potency than the hit compound, exhibiting an *IC*_50_ value of 0.501 μM. Interestingly, introducing a halogen (such as Br, Cl, or F) at the *p*-position of acetanilide led to the generation of compounds **9**–**11**, which show excellent inhibition of CDK9.

Furthermore, introducing bromine at this position, as in the formation of compound **9**, seemed to be more favorable to the inhibitory activity compared with the introduction of other halogens at the same position, with compound **9** showing an *IC*_50_ value of 0.131 μM. Analogously, increasing the lipophilicity of the benzylacetamide substituents in compounds **12**–**14** could be essential for improving their CDK9 inhibitory activity compared to their corresponding acetanilide-containing counterparts, such as compounds **6** and **11**. It was observed that the phenylpropanamide substituents in compounds **15** and **16**, which exhibit *IC*_50_ values of 0.444 and 0.350 μM, respectively, were not conducive to improving their CDK9 inhibitory activity when compared to the corresponding phenylacetanilide substituents in compounds **3** and **10**, which have *IC*_50_ values of 0.421 and 0.193 μM. Among the investigated compounds, compounds **7** and **9** showed the best CDK9 inhibition activities, with *IC*_50_ values of 0.115 and 0.131 μM, respectively.

Compounds **17**–**20** and **23**–**29** were synthesized to investigate their SAR further (Table 2). Compounds **17**–**20** are 4-quinazolinone derivatives with different substituents at 2 and 6 positions. Keeping position 2 unsubstituted while introducing iodine at position 6 resulted in the formation of compound **17**, which showed sub-micromolar activity against CDK9, exhibiting an *IC*_50_ value of 0.639 μM. Keeping the iodine at position 6 while introducing a thiophene ring at position 2, as in compound **18**, led to a doubling of the potency of the compound, with it exhibiting an *IC*_50_ value of 0.296 μM. Introducing *p*-tolyl and chlorine groups at positions 2 and 6, respectively, leading to the formation of compound **19**, did not lead to any significant improvement in the CDK9 inhibitory activity, with the compound showing an *IC*_50_ value of 0.282 μM. However, replacing the *p*-tolyl group at position 2 with a *p*-chlorophenyl group and the chlorine with a nitro group, which is firmly electron withdrawing, led to the formation of compound **20**, which exhibited a 1-fold reduction in potency against CDK9. Five bromophenyl derivatives with a methyl group at position 6 were synthesized to further explore the compounds’ SAR. The quinazolin-4-one derivative (**25**), with a (3-bromophenyl) moiety at position 2, showed significant improvement in CDK9 inhibitory activity, whereas the quinazoline-4-thione analog **26** exhibited a 2-fold reduction in inhibitory activity, with *IC*_50_ values of 0.142 and 0.289 μM, respectively. Replacing 4-quinazolinone with benzo[d][1,3]oxazin-4-one, as in compound **24**, led to a 3-fold reduction in CDK9 inhibitory activity compared with compound **25**. Interestingly, intermediate **23**, used in the synthesis of compound **24**, was tested and shown to be more potent, with *IC*_50_ values of 0.210 and 0.486 μM, respectively. *N*-hydroxy derivative **28** was shown to be a less potent inhibitor of CDK9 when compared to compound **25**, showing *IC*_50_ values of 0.210 and 0.142 μM, respectively. Compound **27**, the *N*-amino analog of compound **25**, showed an approximately 6-fold decrease in potency against CDK9. A quinazoline-2,4-dione analog, compound **29**, was also tested and exhibited inhibitory activity against CDK9, with an *IC*_50_ value of 0.589 μM.

Breast cancer often occurs with dysregulation in CDK9 levels, and several studies have shown the efficacy of CDK9 inhibitors in breast cancer [20,21]. Therefore, the antiproliferative activity of compounds **1**–**20** and **23**-**29** was evaluated against the breast cancer cell line MCF-7 by the metabolic assay MTT. Compounds **1**–**16** showed potent cytotoxic activities with *IC*_50_ values ranging from 0.16 to 4.65 μM, with compounds **4** and **5** being the most potent cytotoxic agents, which could be due to their multitarget inhibitory activities against EGFR, HER2, and VEGFR2 as well as CDK9 [26]. However, compounds **17**–**20** and **23**–**29** were significantly less potent than compounds **1**–**16,** with *IC*_50_ values ranging from 3.88 to 28.7 μM.

### 2.3. Molecular Docking

Molecular docking experiments were performed using the genetic algorithm docking program GOLD 5.2 to rationalize the observed potency of compounds **7**, **9**, and **25**. In addition, flavopiridol was used as a reference compound to compare the binding pattern. The modeled complexes with CDK9 are shown in Figure 4.

At the binding cavity of CDK9, flavopiridol, compounds **7**, **9**, and **25** occupy the adenosine triphosphate (ATP) binding site. Compounds **7** and **9** display similar binding modes. These conformations, however, are different from the ones shown by flavopiridol. The benzene ring of the quinazolinone is in contact with the Phe105 residue in the hinge region. In addition, the sulphonamide moieties of compounds **7** and **9** bind differently, forming hydrogen bonds with the Thr29 and Glu107 residues in the enzyme, respectively.

Moreover, the Ethoxy group of compound **7** forms a hydrogen bond with Lys48, whereas there is no interaction between the enzyme and the anilide moiety of compound **9**. This suggests that the sulphonamide moiety is essential for the activity, whereas the acetanilide groups could be necessary for the potency in the context of 2-mercapto-3-(4-sulfamoylphenethyl)quinazolin-4-one. However, compound **25** adopts a similar orientation to that of flavopiridol. Unlike compounds **7** and **9**, the NH group of compound **25** forms a hydrogen bond with Glu107. Moreover, the methyl group of compound **25** is in contact with Phe103 of the gatekeeper region of the enzyme. In addition, the m-bromophenyl group in compound **25** occupies a similar position to that occupied by the *o*-chlorophenyl ring of flavopiridol.

### 2.4. Analyses of the Physicochemical Properties of the Compounds

#### 2.4.1. Lipinski’s Rule of Five

Lipinski’s rule of five was used to evaluate the drug-like properties of compounds **1**–**20** and **23**–**29**. DataWarrior was used to estimate the molecular weight (MW), *CLogP*, hydrogen bond acceptors (HBAs), and hydrogen bond donors (HBD) for each molecule, and the values are presented in Table 3. The data in the table show that compounds **4**–**16** have molecular weights of >500 Da. In addition, compound **8** offers an additional violation of Lipinski’s rule, with HBAs >10. However, all the compounds satisfy Lipinski’s rule regarding lipophilicity and the number of HBDs, with *CLogP* and HBD values below 5.

#### 2.4.2. Ligand Efficiency (LE)

The *LE* is a property that describes the potency per heavy atom of a drug [50,51,52]. The *LE* values of the synthesized compounds were obtained using DataWarrior according to the following equation [50,52]:
(1)$$LE=\frac{-RT\text{ln}IC_{50}}{N}$$


*N* represents the number of heavy atoms, i.e., non-hydrogen atoms in the drug, *R* is the universal gas constant, *T* is the absolute temperature in degrees Kelvin, and *IC*_50_ is CDK9 *IC*_50_ in mol/L.

*LE* is an essential metric in lead optimization, which allows the comparison of the affinity of molecules according to their size. Compounds with *LE* values higher than 0.3 are considered promising lead compounds. The *LE* values for the target compounds in this study are presented in Table 4, which shows that the *LE* values of the synthesized compounds are between 0.2 and 0.7. Except for compounds **3**–**16**, the *LE* values fall into an acceptable range and are >0.3 [52,53].

#### 2.4.3. Ligand Lipophilic Efficiency (LLE)

*LLE* is used to link the potency of a compound to its lipophilicity [52,53]. The challenge in drug discovery is optimizing a compound’s activity while maintaining lipophilicity at a constant value. For this reason, *LLE* is considered an effective strategy to control the lipophilicity of a molecule to avoid any “molecular obesity” during lead optimization. The *LLE* values for compounds **1–20** and **23–29** shown in Table 4 were obtained using DataWarrior according to the following equation [52]:
(2)$$LLE=pIC_{50}-\;CLogP$$


*p**IC*_50_ is the negative log of the CDK9 *IC*_50_ and *CLogP* is the calculated LogP value.

An acceptable lead compound should have an *LLE* value of ≥ 5. Compounds **2**, **17**, and **27** show good *LLE* values, i.e., *LLE* > 5 [52,53]. However, the other compounds have values that are below the recommended limit.

In conclusion, compound **17** is a good candidate for lead optimization since it has the lowest non-hydrogen atoms (*N*), an acceptable *LE* value of 0.708, and an acceptable *LLE* value of 5.20.

## 3. Materials and Methods

### 3.1. Chemistry

Chemicals and solvents were obtained from suppliers and used directly without any purification. Agilent 6320 Ion Trap mass spectrometer was used to generate mass spectra (MS). Melting Point Apparatus Barnstead 9100 Electrothermal was used to record the final compounds’ melting points (uncorrected). IR spectra were obtained using an FT-IR Perkin-Elmer spectrometer. Bruker 700 Ultrashield NMR spectrometer was run at 700 MHz and 175 MHz to generate ^1^H and ^13^C spectra, respectively. Compounds **1**–**20** and **29** were synthesized as reported previously [22,27,40,41,42,43]. the newly synthesized compounds were re-crystalized from ethanol. IR, NMR and mass spectra of compounds **23–28** are available in the Appendix A of this article.

*2-(3-Bromobenzamido)-5-methylbenzoic acid* (**23**). Equimolar amounts of 2-amino-5-methylbenzoic acid (20 mmol, 3.0 g) and 3-bromobenzoyl chloride (20 mmol, 4.40 g) were stirred at room temperature in 20 mL of anhydrous pyridine for 3 h. The solvent was then removed *in vacuo*, and the resulting crude solid was washed with 5% HCl ice–water before being filtered and dried to give the final product in 97% yield. M.P. 200–202 °C; IR (KBr, cm^−1^) ν: 3233 (NH), 2600 (COOH), 1653 (C=O); ^1^H NMR (DMSO-d6): δ 12.06 (s, ^1^H), 8.51 (s, ^1^H), 8.50 (d, ^1^H, J = 8.47 Hz), 8.08 (s, ^1^H), 7.93 (d, ^1^H, *J* = 7.70 Hz), 7.84 (t, 2H, *J* = 6.79 and 7.98 Hz), 7.55 (t, ^1^H, *J* = 7.84 Hz), 7.47 (d, ^1^H, *J* = 8.47 Hz), 2.33 (s, 3H); ^13^C NMR (DMSO-d6): δ 170.52, 163.37, 138.74, 137.27, 135.22, 135.20, 132.94, 131.73, 131.64, 130.26, 126.49, 122.64, 120.65, 117.46, 20.73; MS [M–OH; 316 and M–OH + 2: 318; M–COOH; 288 and 290].

*2-(3-Bromophenyl)-6-methyl-4H-benzo[d][1,3]oxazin-4-one* (**24**). 2-(3-Bromobenzamido)-5-methylbenzoic acid (**23**) (15 mmol, 4.74 g) was boiled in acetic anhydride for 4 h, after which the reaction mixture was cooled, and the solid was collected by filtration and dried to give the final product in 95% yield. M.P. 155–157 °C; IR (KBr, cm^−1^) ν: 1753 (C=O); ^1^H NMR (DMSO-d6): δ 8.20 (s, ^1^H), 8.11 (d, ^1^H, *J* = 7.77 Hz), 7.92 (s, ^1^H), 7.83 (d, ^1^H, *J* = 7.77 Hz), 7.75 (d, ^1^H, *J* = 8.12 Hz), 7.60 (d, ^1^H, *J* = 8.12 Hz), 7.54 (t, ^1^H, *J* = 7.91 and 7.84 Hz), 2.45 (s, 3H); ^13^C NMR (DMSO-d6): δ 159.02, 154.70, 144.23, 139.53, 138.39, 135.57, 132.78,131.66, 130.37, 128.05, 127.35, 127.02, 122.55, 117.12, 21.22; MS [*m/z*: 315 and M + 2: 317].

*2-(3-Bromophenyl)-6-methylquinazolin-4(3H)-one* (**25**). 2-(3-Bromophenyl)-6-methyl-4H-benzo[d][1,3]oxazin-4-one (**24**) (5 mmol 1.58 g) was heated in formamide (7 mL) for 10 h, after which the reaction mixture was cooled, and the resulting solid was filtered and dried to give the final product in 80% yield. M.P. 320–322 °C; IR (KBr, cm^−1^) ν: 3334 (NH), 1660, (C=O); ^1^H NMR (DMSO-d6): δ 8.35 (s, ^1^H), 8.15 (d, ^1^H, *J* = 7.00 Hz), 7.95 (s, ^1^H), 7.78 (d, ^1^H, *J* = 7.211 Hz), 7.68 (s, 2H), 7.51 (t, ^1^H, *J* = 7.28 and 7.35 Hz), 2.46 (s, 3H); ^13^C NMR (DMSO-d6): δ 165.30, 162.66, 150.70, 146.91, 137.16, 136.41, 135.55, 134.33, 131.22, 130.74, 127.88, 127.13, 125.72, 122.35, 21.34; MS [*m/z*: 314 and M + 2: 316].

*2-(3-Bromophenyl)-6-methylquinazoline-4(3H)-thione* (**26**). 2-(3-Bromophenyl)-6-methylquinazolin-4(3H)-one (**25**) (3 mmol, 945 mg) was heated with phosphorus pentasulfide in dry toluene (5 mL) for 12 h, after which time the reaction mixture was cooled, and the solid obtained was filtered and dried to give the final product in 68% yield. M.P. 259–260 °C; IR (KBr, cm^−1^) ν: 3102 (NH), 1241 (C=S); ^1^H NMR (DMSO-d6): δ 13.94 (s, ^1^H), 8.40 (s, ^1^H), 8.34 (s, ^1^H), 8.15 (d, ^1^H, *J* = 7.84 Hz), 7.79 (d, ^1^H, *J* = 7.91 Hz), 7.74 (dd, ^1^H, *J* = 1.33 Hz), 7.70 (d, ^1^H, *J* = 8.26 Hz), 7.51 (t, ^1^H, *J* = 7.91 Hz); 2.49 (s, 3H); ^13^C NMR (DMSO-d6): δ 187.62, 149.86, 142.67, 138.70, 137.49, 134.78, 134.45, 131.42, 131.05, 128.94, 128.66, 127.94, 127.89, 122.13, 21.54; MS [*m/z*: 330 and M + 2: 332].

*3-Amino-2-(3-bromophenyl)-6-methylquinazolin-4(3H)-one* (**27**). 2-(3-Bromophenyl)-6-methyl-4H-benzo[d][1,3]oxazin-4-one (**24**) (5 mmol 1.58 g) was heated with absolute hydrazine hydrate (3 mL) and absolute ethanol (3 mL) for 8 h, after which time the reaction mixture was cooled. The separated solid was filtered and dried to give the final product an 87% yield. M.P. 170–172 °C; IR (KBr, cm^−1^) ν: 3401, 3308 (NH2), 1668 (C=O); ^1^H NMR (DMSO-d6): δ 8.00 (t, ^1^H, *J* = 1.68 and 1.75 Hz), 7.99 (s, ^1^H), 7.81 (d, ^1^H, *J* = 7.84 Hz), 7.70 (tt, ^1^H, *J* = 0.98 and 0.77 Hz), 7.67 (dd, ^1^H, *J* = 1.82 Hz), 7.64 (d, ^1^H, *J* = 8.119 Hz), 7.45 (t, ^1^H, *J* = 7.91), 5.66 (s, 2H), 2.48 (s, 3H); ^13^C NMR (DMSO-d6): δ 161.53, 154.06, 145.12, 137.56, 137.28, 136.21, 132.66, 132.60, 130.09, 129.16, 127.83, 125.72, 120.96, 120.47, 21.35; MS [*m/z*: 329 and M + 2: 331]. 

*2-(3-Bromophenyl)-3-hydroxy-6-methylquinazolin-4(3H)-one* (**28**). 2-(3-Bromophenyl)-6-methyl-4H-benzo[d][1,3]oxazin-4-one (**24**) (5 mmol 1.58 g) was heated with hydroxylamine hydrochloride (6 mmol, 417 mg) in dry pyridine (10 mL) for 15 h. The reaction mixture was cooled, and the solvent was removed *in a vacuo*. The solid obtained was then washed with 5% HCl ice–water and filtered, dried, and recrystallized from ethanol to give the final product an 84% yield. M.P. 235–237 °C; IR (KBr, cm^−1^) ν: 3301(OH), 1670 (C=O); ^1^H NMR (DMSO-d6): δ 11.73 (s, ^1^H), 9.00 (d, 2H, *J* = 9.24 Hz), 7.84 (d, ^1^H, *J* = 7.77 Hz), 7.76 (dd, ^1^H, *J* = 0.84 and 0.92 Hz), 7.68 (q, 2H, *J* = 8.19 and 8.26 Hz), 7.49 (t, ^1^H, *J* = 7.91 Hz), 2.48 (s, 3H); ^13^C NMR (DMSO-d6): δ 158.55, 151.48, 144.59, 137.43, 136.12, 135.47, 133.39, 132.41, 130.54, 128.97, 127.99, 125.64, 121.81, 121.36, 21.33; MS [M–OH: 313 and M–OH + 2: 315].

### 3.2. Metabolic Assay

The antiproliferative activity of the twenty-seven compounds was evaluated by 3-(4,5-dimethylthiazol-2-yl)-2,5-diphenyltetrazolium bromide (MTT) metabolic assay against the MCF-7 cell line, according to a previous method [26].

### 3.3. CDK9 Kinase Assay

In vitro luminescent CDK9 kinase assay was performed as reported previously using Kinase- Glo^®^ MAX as a detection reagent [43]. Briefly, 5 μL of each inhibitor in concentrations ranging from 10 μM to 1 nM (10 μM, 1 μM, 0.1 μM, 0.01 μM, and 0.001 μM) and 10 μL of enzyme substrate were mixed in 20 μL of kinase assay buffer (obtained from BPS Bioscience, catalog #79334) at room temperature. Then 20 μL of 5 ng/μL CDK9/cyclin T was added to the mixture to initiate the reaction. After 45 min, 50 μL of Kinase-Glo^®^ Max reagent was added, and the resulting mixture was incubated for 15 min at room temperature. The chemiluminescence was measured microplate reader, and *IC*_50_ values were calculated using Prism 8.0 (GraphPad Software, San Diego, CA, USA).

### 3.4. Molecular Docking

Molecular docking was performed according to the procedure reported previously using the X-ray crystal structure of flavopiridol in a complex with CDK9 (PDB ID: 3BLR) which was retrieved from the PDB Data Bank (URL: http://www.rcsb.org; accessed on 20 September 2022) [26].

## 4. Conclusions

Twenty-seven compounds were synthesized, and their CDK9 inhibitory and cytotoxic activities were evaluated. Compounds **7**, **9**, and **25** were the most potent CDK9 inhibitors, with *IC*_50_ values of 0.115, 0.131, and 0.142 μM, respectively. The binding modes of these molecules were studied using molecular docking, which showed that they occupy the ATP binding site of CDK9. Of these three molecules, compound **25** shows good drug-like properties since it does not violate Lipinski’s rule of five. In addition, this molecule shows promising *LE* and *LLE* values and is an ideal candidate for further optimization.

## Data Availability

Data are available by request from the corresponding author.

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
