# Peer review of "Synthesis, Cytotoxic Evaluation, and Structure-Activity Relationship of Substituted Quinazolinones as Cyclin-Dependent Kinase 9 Inhibitors"

_molecules, 2022, doi:10.3390/molecules28010120_

Round 1
Reviewer 1 Report
This manuscript projects an overview of substituted quinazolines as CDK9 inhibitors and their synthesis, cytotoxcity and SAR evaluations. Author reported 29 molecules and some of them shown most potent CDK9 inhibition. As well as performed molecular docking reveals Adenosine triphosphate binding site of CDK9. In my opinion, this manuscript suit for molecules criteria and some issues need to be fixed before its acceptence.
comments
1.compound numbering should not be given in the abstract.
2.CMGC means ???
3. compound 29appeared just after 20 need to adressed.
4. yields should be mentioned in scheme 2 and does it that much space for scheme??
5. line 154, compound nos should be bold .
6. SI file needs to be improved better, in all spectra compound structure as well as with detailed information of solvent, nuclei etc. NMR data should be rechecked again and NMR window should follow 0-10 ppm for 1H and 0-210 ppm for 13C.
Author Response
- Compound numbering should not be given in the abstract.
- It was changed accordingly.
- CMGC means ???
- It means [Cyclin-dependent kinases (CDKs), Mitogen-activated protein kinases (MAPKs), Glycogen synthase kinases (GSKs), and Cdc2-like kinases (CLKs)], which is one of the protein kinases superfamilies.
- Compound 29appeared just after 20 need to addressed.
- The synthesized compounds for CDK-9 were 27 compounds and corrected throughout the manuscript
- Yields should be mentioned in scheme 2 and does it that much space for scheme??
- It was fixed
- Line 154, compound nos should be bold.
- It was changed accordingly.
- SI file needs to be improved better, in all spectra compound structure as well as with detailed information of solvent, nuclei etc. NMR data should be rechecked again and NMR window should follow 0-10 ppm for 1H and 0-210 ppm for 13
- It was Fixed, and the NMR spectrometer was run at 700 MHz and 175 MHz to generate 1H and 13C spectra, respectively, as mentioned in Materials and Methods (Chemistry).
Reviewer 2 Report
· Add detailed methodology for cytotoxicity studies and cell line culturing in 2 separate sections. What concentrations of the compounds were tested, how many concentrations, in how many repeats, in how many experiments, what was the exposure time…etc.
· Where the cell line originated from, which passage it was from, how it was cultured, passaged, cell viability assessed before the experiment... etc.
· Why MCF-7 cell line was selected for the study, justify please.
· What is the standard error of MTT assay in your laboratory?
· Give detailed methodology to section 4.3.
· MTT- is not test for antiproliferative activity, it should be corrected in the whole manuscript, it is metabolic assay.
· There is no statistical analysis. It should be added.
· What is applicable value of the research? It should be discussed.
· Provide the limitations and strengths of your study.
Author Response
- Where the cell line originated from, which passage it was from, how it was cultured, passaged, cell viability assessed before the experiment... etc.
- The cells were obtained from the cell bank in the College of Pharmacy at King Saud University. The culturing method and experiment conditions were explained extensively in our previous work in reference # 26.
- Why MCF-7 cell line was selected for the study, justify please.
- “Breast cancer often has dysregulation in CDK9 levels, and several studies have shown the efficacy of CDK9 inhibitors in breast cancer [20, 21]”. This was justified in the introduction and discussion.
- What is the standard error of MTT assay in your laboratory?
- The SD and SE are usually variable between experiments in our lab; however, both are generally valid compared to the mean. The values of SD and SE are usually less than the mean after multiplication by 2.5 and 10, respectively.
- Give detailed methodology to section 4.3.
- A detailed methodology was added.
- MTT- is not test for antiproliferative activity, it should be corrected in the whole manuscript, it is metabolic assay.
- It was changed accordingly in the entire manuscript.
- There is no statistical analysis. It should be added.
- Statistical analysis has been added in the footnotes of Tables 1 and 2.
Reviewer 3 Report
Reviewers’ comments for the Manuscript ID: 2056740
The manuscript title: “Synthesis, cytotoxic evaluation, and structure–activity relationship of substituted quinazolinones as cyclin-dependent kinase 9 inhibitors”,
In the current manuscript authors evaluated the CDK9 inhibitory and cytotoxic activities and molecular docking studies of recently published Dual inhibitors of EGFR/HER2. Among all the tested molecules compounds 7, 9, and 25 were found to be most potent CDK9 inhibitors, with IC50 values of 0.115, 0.131, and 0.142 mM, respectively. Given importance of topic, these molecules could be potentially useful for further design of potent CDK9 inhibitors, so the manuscript is suitable for publication in “Molecules” after addressing few general comments.
General comments:
1) Introduction part is very brief and not clearly written about significance of the topic, several most potent CDK9 inhibitors have been reported and many are in different stages of clinical development, authors need to elaborate the introduction part.
2) Whole manuscript needs to be correct grammatically,
3) Some compounds numbers in the text are bold in style, some are not, it need to be make uniform.
4) IC50 of compound 7 and 9 in line 159 needs to be correct from 0.062 and 0.075 μM to 0.115 and 0.131 uM respectively.
Author Response
- Introduction part is very brief and not clearly written about significance of the topic, several most potent CDK9 inhibitors have been reported and many are in different stages of clinical development, authors need to elaborate the introduction part.
- The introduction was elaborated accordingly
- Whole manuscript needs to be correct grammatically
- The manuscript was sent for language editing, and the certificate is attached.
- Some compounds numbers in the text are bold in style, some are not, it need to be make uniform.
- This was fixed in the entire manuscript.
- IC50 of compound 7 and 9 in line 159 needs to be correct from 0.062 and 0.075 μM to 0.115 and 0.131 uM respectively.
- This was fixed.
Reviewer 4 Report
Herein the authors have synthesized 6-new compounds besides 23 known compounds.
Compound 1. Ref 40, with subnanomolar carbonic anhydrase II and XII inhibitory properties
Compounds (2-16). Ref 22, with Dual inhibitors of EGFR/HER2
Compounds 17-20 and 29. Ref 27, 41, 43, with anticonvulsant activity, Poly(ADP-ribose) polymerase-1 inhibitors, and receptor 2 (PAR(2)) antagonists
The new compounds were tested along with known compounds for detection of a cyclin-dependent kinase 9 inhibitor.
Overall work is good, but there are some notes
Introduction
1. Page 2\16, Lines 60. Please refer to your previous work (insert ref) as you mentioned at the beginning of the sentence.
2. Page 3\16, Lines 72-74. Has no sense after the previous explanation (lines 60-69) or the next one at results and discussion
Results and discussion
1. The interpretation of spectroscopic characterization of the new compounds, Pages 4 and 5, Lines 91-119 not properly written. The interpretation needs to be re-written to be simple and clear.
2. Table 1 and 2. Authors have to write that, Results are represented as the mean of three independent experiments (n=3) or the data are expressed as mean ±SD.
3. Molecular docking. Page 9\16. The 3D binding mode of the studied compounds as illustrated in Figure 4 not clear and needs to be re-shot to clear the key amino acids of the CDK9 active site as well as the formed hydrogen bonds with the studied compounds
4. Analyses of the physicochemical properties of the compounds, Page 10\16. What program did the authors use to estimate the Lipinski judgment? Is it DataWarrior? If it is, please indicate it with a reference.
Materials and Methods
1. Mass spectra interpretation of compounds that contain bromine atoms should be written as M\M+2 (1:1). For example compound 24, MS [m/z: M\M+s (315 \ 317) (1:1)]. As the exact molecular weight is 316 due to the Br atomic weight is 80. The Br atom in the mass spectrum is calculated according to its isotopes 79\81
2. Are the new compounds 23, 24, 25, 26, and 27 purified by crystallization to ensure that you got pure target compounds?! If this is done, where is the solvent of crystallization?
Author Response
Introduction
- Page 2\16, Lines 60. Please refer to your previous work (insert ref) as you mentioned at the beginning of the sentence.
- The reference was introduced.
- Page 3\16, Lines 72-74. Has no sense after the previous explanation (lines 60-69) or the next one at results and discussion
- The mentioned sentence was removed
Results and discussion
- The interpretation of spectroscopic characterization of the new compounds, Pages 4 and 5, Lines 91-119 not properly written. The interpretation needs to be rewritten to be simple and clear.
- The discussion was rewritten with yellow highlights in the manuscript.
- Table 1 and 2. Authors have to write that, Results are represented as the mean of three independent experiments (n=3) or the data are expressed as mean ±SD.
- The sentence “Data are expressed as mean ± SD” has been added in the footnotes of Tables 1 and 2.
- Molecular docking. Page 9\16. The 3D binding mode of the studied compounds as illustrated in Figure 4 not clear and needs to be re-shot to clear the key amino acids of the CDK9 active site as well as the formed hydrogen bonds with the studied compounds
- The figures are updated accordingly.
- Analyses of the physicochemical properties of the compounds, Page 10\16. What program did the authors use to estimate the Lipinski judgment? Is it DataWarrior? If it is, please indicate it with a reference.
- It was mentioned in section 2.4.
Materials and Methods
- Mass spectra interpretation of compounds that contain bromine atoms should be written as M\M+2 (1:1). For example compound 24, MS [m/z: M\M+s (315 \ 317) (1:1)]. As the exact molecular weight is 316 due to the Br atomic weight is 80. The Br atom in the mass spectrum is calculated according to its isotopes 79\81
- The Mass has been corrected accordingly.
- Are the new compounds 23, 24, 25, 26, and 27 purified by crystallization to ensure that you got pure target compounds?! If this is done, where is the solvent of crystallization?
- The newly synthesized compounds were re-crystalized from ethanol (edited in Materials and Methods, Chemistry)
Round 2
Reviewer 1 Report
The suggested remarks have been made by the authors and recommended for acceptance.
Reviewer 2 Report
I have no more comnents.